Zoledronic acid promotes osteoclasts ferroptosis by inhibiting FBXO9-mediated p53 ubiquitination and degradation

Qu Xingzhou
Sun Zhaoqi
Wang Yang 115078@sh9hospital.org.cn
Ong Hui Shan 117069@sh9hospital.org.cn
Department of Oral and Maxillofacial-Head & Neck Oncology, Ninth People’s Hospital Affiliated to Shanghai Jiao Tong University School of Medicine , SH, Shanghai , China
Felley-Bosco Emanuela
Electronic publication date: 2021 Dec 16
Publication date: 2021
Volume: 9
Electronic Location ID: e12510
Received 2021 Jul 7; Accepted 2021 Oct 27
Copyright: © 2021 Qu et al.
Copyright year: 2021
Copyright holder: Qu et al.
License: This is an open access article distributed under the terms of the Creative Commons Attribution License, which permits unrestricted use, distribution, reproduction and adaptation in any medium and for any purpose provided that it is properly attributed. For attribution, the original author(s), title, publication source (PeerJ) and either DOI or URL of the article must be cited.
License URL: https://creativecommons.org/licenses/by/4.0/

Keywords: Bisphosphonates related osteonecrosis of jaw, Zoledronic acid, Ferroptosis, FBXO9, p53

Funding: The authors received no funding for this work.

==============================
Bisphosphonates (BPs)-related osteonecrosis of jaw (BRONJ) is a severe complication of the long-term administration of BPs. The development of BRONJ is associated with the cell death of osteoclasts, but the underlying mechanism remains unclear. In the current study, the role of Zoledronic acid (ZA), a kind of bisphosphonates, in suppressing the growth of osteoclasts was investigated and its underlying mechanism was explored. The role of ZA in regulating osteoclasts function was evaluated in the RANKL-induced cell model. Cell viability was assessed by cell counting kit-8 (CCK-8) assay and fluorescein diacetate (FDA)-staining. We confirmed that ZA treatment suppressed cell viability of osteoclasts. Furthermore, ZA treatment led to osteoclasts death by facilitating osteoclasts ferroptosis, as evidenced by increased Fe2+, ROS, and malonyldialdehyde (MDA) level, and decreased glutathione peroxidase 4 (GPX4) and glutathione (GSH) level. Next, the gene expression profiles of alendronate- and risedronate-treated osteoclasts were obtained from Gene Expression Omnibus (GEO) dataset, and 18 differentially expressed genes were identified using venn diagram analysis. Among these 18 genes, the expression of F-box protein 9 (FBXO9) was inhibited by ZA treatment. Knockdown of FBXO9 resulted in osteoclasts ferroptosis. More important, FBXO9 overexpression repressed the effect of ZA on regulating osteoclasts ferroptosis. Mechanistically, FBXO9 interacted with p53 and decreased the protein stability of p53. Collectively, our study showed that ZA induced osteoclast cells ferroptosis by triggering FBXO9-mediated p53 ubiquitination and degradation.

Introduction

Bisphosphonates (BPs) inhibit osteoclast activity and disrupt osteoclast mediated bone resorption (Zhang, Gangal & Uludag, 2007; Black et al., 2019). BPs are widely used in the treatment of bone metastasis cancer (La-Beck et al., 2021), osteoporosis (Favus, 2010), and multiple myeloma (Mhaskar et al., 2017). BRONJ is an injury of the jaw that affects patients treated with BPs. Since it was reported in 2003, BRONJ has been considered as a common and important adverse side effect of BPs treatment, especially nitrogen-containing bisphosphonates (Residronate, Alendronate, and Zoledronic acid (ZA)) (Marx, 2003). There are various hypotheses for the development of BRONJ, the most recognized hypothesis was bone remodeling suppression. Although significant progress has been made in prevention and treatment of BRONJ base on the hypotheses, the mechanisms underlying the development of BRONJ remain unclear.

Osteoclasts, members of the monocyte/macrophage hematopoietic, play an important role in the progression of bone remodeling (Hadjidakis & Androulakis, 2006). RAW264.7 cells and bone marrow-derived macrophages (BMDMs) can be induced into osteoclasts by receptor activator of nuclear factor-κB ligand (RANKL), and has been widely used as a cell model for the study of osteoclast related diseases in vitro (Ikebuchi et al., 2018). The number and the resorptive function of osteoclasts were usually increased during the process of bone remodeling (Madel et al., 2019). Therefore, osteoclast is one of the core targets for the treatment of osteoporosis and other bone-remodeling-related diseases. It is well known that ZA can lead to a stronger inhibition of osteoclasts differentiation and induces the apoptosis of osteoclasts (Wang et al., 2020), while the underlying mechanism of ZA in the function of osteoclasts reminds unclear.

Ferroptosis is a recently identified type of iron-mediated cell death. Unlike other forms of programmed cell death, such as apoptosis and necroptosis, ferroptosis does not involve the activation of caspase protein (Li et al., 2020; Hirschhorn & Stockwell, 2019). It is characterized by an increased level of lipid peroxidation products and reactive oxygen species (ROS). The dysregulation of ferroptosis has been related to many pathological processes, such as cancer (Mou et al., 2019), neurodegenerative diseases (Reichert et al., 2020), and inflammation-related diseases (Sun et al., 2020). More and more studies showed that ferroptosis contribute to the development of BRONJ. Bagan et al. (2014) found that the levels of MDA, GSSG, and 8-oxo-dG and the GSSG/GSH ratio in serum and saliva were significantly higher in patients with BRONJ compared with controls. Ma et al. (2020) demonstrated that melatonin suppresses osteoblast ferroptosis and improved the osteogenic capacity of MC3T3-E1 by activating the Nrf2/HO-1 pathway. However, whether ferroptosis was involved in the osteoclasts differentiation and death induced by ZA is still unknown. In the current study, we showed that ZA inhibits the osteoclasts viability in a dose-dependent manner. For the first time, we showed that ZA promotes the ferroptosis of osteoclast by increasing the protein stability of p53. ZA-induced downregulation of ubiquitin E3 ligase FBXO9, and FBXO9 overexpression restores cell viability inhibition of osteoclast induced by ZA. Moreover, FBXO9 facilitates ubiquitination-mediated degradation of p53.

Materials and Methods

Cell culture

RAW264.7 cells were purchased from the ATCC (TIB-71; Manassas, VA, USA) and cultured in alpha-Modified Eagle’s Medium (α-MEM; Gibco, Jenks, OK, USA) with 100 U/ml penicillin, and 100 μg/ml streptomycin at 37 °C in a humidified atmosphere containing 5% CO2 and 95% air. To BMDMs, bone marrow cells were purchased from the ATCC (CRL-2420; Manassas, VA, USA) and cultured in RPMI 1640 medium (Thermo Fisher Scientific, Waltham, MA, USA) for 6 days with 100 U/ml penicillin, 100 μg/ml streptomycin, 10% FBS, and 10 ng/mL recombinant mouse macrophage colony-stimulating factor (PeproTech, Cranbury, NJ, USA). For osteoclast formation assay.RAW264.7 cells were seeded in 12-well plate s (1 × 104 cells/well) supplemented with 50 ng/ml RANKL (R&D Systems) for 6 days. BMDMs (1 × 104 cells/well) were cultured in the presence of M-CSF (10 ng/mL) and RANKL (50 ng/mL) for 6 days.

TRAP staining

TRAP histochemical staining was performed to confirm the osteoclast as previously described (Takahashi et al., 2003), by using an acid phosphatase, leukocyte (TRAP) kit (Sigma-Aldrich, St. Louis, MO, USA). Brifely, 1 × 105 BMDMs or RAW264.7 induced osteoclasts were then fixed in 10% neutral-buffered formalin (NBF) solution for 20 min, then the NBF solution was replaced with TRAP staining solution and 0.1% Fast Red AS TR salt at RT. After 45 min cells were washed with 1× PBS for 3 times and imaged.

Measurement of cell viability

Cell viability of BMDMs and RAW264.7 was assessed by using a CCK-8 and FDA assay (Dojindo, Kumamoto, Japan). For CCK-8 assay, BMDMs and RAW264.7 cells (1 × 104 cells per well) were seeded in 96-well plates for 24 h, then cells were treated with different doses of ZA (5, 10, and 50µM) for 48 h. To analyze the cells death of osteoclast, BMDMs, and RAW264.7 (5 × 103 cells per well) were treated with 10 μM of ZVAD-FMK, 2 μM of Fer-1, or 10 μM of necrostatin-1, for 48 h with or without ZA (50 μM). Then, a total 10 µL of CCK-8 reagent was added to each well additional 4 h at 37 °C with 5% CO2, and the absorbance at 450 nm of each well was assayed using a microplate reader (BioTek Instruments). For FDA assay, after culture with different dose ZA (5, 10, and 50 µM) for 48 h, BMDMs and RAW264.7 cells were treated with 10 μl of FDA solution (5 mg/mL; Invitrogen, Waltham, CA, USA) at 37 °C with 5% CO2 for 20 min, then cells images were obtained using a fluorescence microscope (Olympus Corporation, Tokyo, Japan).

Fe2+ concentration

The concentration of Ferrous iron (Fe2+) in BMDMs and RAW264.7 cells in the presence or absence of ZA (50 μM) was assessed using an iron assay kit (MAK025; Sigma-Aldrich, St. Louis, MO, USA) as the manufacturer’s instructions. Briefly, cell samples were incubated with 10 μL of iron reducer for 30 min at RT, then 100 μL iron probe was added to trigger the reaction; thus, the absorbance was measured at 593 nm.

Lipid reactive oxygen species (ROS) assay

Lipid ROS level in BMDMs and RAW264.7 cells in the presence or absence of ZA (50 μM) was assayed using C11-BODIPY (Invitrogen, Waltham, MA, USA), a fluorescent-labelled oxidation sensitive probe. In brief, BMDMs and RAW264.7 cells were seeded in 24-well plates (5 × 105/well) and treated with ZA (50 μM) with or without FBXO9 overexpression for 48 h, then BMDMs and RAW264.7 cells were cultured with C11-BODIPY probe with a final concentration of 1 μM in at 37 °C with 5% CO2 for 30 min, then the Lipid ROS levels were assayed using flow cytometer.

MDA and GSH content

MDA in BMDMs and RAW264.7 cells was analyzed using a lipid peroxidation assay kit (ab118970; Abcam, Cambridge, England) in the presence or absence of ZA (50 μM). GSH content in BMDMs and RAW264.7 cells was assayed using a Glutathione Assay Kit (ab65322; Abcam, Cambridge, England) according to the standard protocol. Briefly, cell supernatant, 5, 5′-dithio-bis 2-nitrobenzoic acid solution and the reagents of kits were mixed together and incubated at RT for 10 min, then NADPH was added into this system to trigger the reaction. The absorbance of 5-thio-2-nitrobenzoic acid was detected at 412 nm.

Transient transfection of FBXO9 or si-FBXO9

The recombinant plasmids pcDNA-FBXO9 containing FBXO9 cDNA were sub-cloned into pcDNA3.1 vector via EcoRV/HindIII sites. To overexpress FBXO9, the pcDNA-FBXO9 was transfected into BMDMs cells using Lipofectamine 2000. FBXO9 Knockdown and transfection were performed according to the manufacturer’s instructions. Briefly, cells were transfected with 10 nM of si-FBXO9 RNA (sense AUCAGAAUGACAAUCUUCCUCU, antisense GGAAGAUUGUCAUUCUGAUGCU) or si-Control RNA (sense CAGUCGCGUUUGCGACUGGUU, antisense CCAGUCG-CAAACGCGACUGUU), and the cell was induced by M-CSF (10 ng/mL) and RANKL (50 ng/mL) for further experiment.

Quantitative real-time PCR (qPCR)

After treatment with 50 μM of ZA for 48 h, total RNA was isolated with Trizol reagent (Sigma-Aldrich, , St. Louis, MO, USA) as instructed by the manufacturer. Reverse transcriptional PCR was carried out using the SMART PCR cDNA Synthesis Kits (Clontech, Mountain View, CA, USA ). qPCR was carried out on ABI 7500 RealTime PCR System (Applied Biosystem, Waltham, MA, USA) with powerup SYBR green Mix (Thermo Fisher, Waltham, MA, USA). The fold changes of RNA transcripts were calculated by the 2−ΔΔCt method and the 18s was used as a reference gene. The qPCR primer pairs in Table 1.

Table 1 Sequence.

CFAP53:	
forward primer:5′-GACAAAATGAGAGAGAGAACCAAGT-3′	
reverse primer: 5′-TCCCTGAACTGCTGGTCTAAC-3′	
COL14A1	
forward primer:5′-ACTGGTTTTCACGGGTGTTC-3′,	
reverse primer: 5′-TAAGTCGAGGAGAGGCAAGC-3′	
ARSJ	
forward primer:5′-CTGAGATAAAGACGCCCACC-3′,	
reverse primer: 5′-ATAGAATGCTGAAGTCCCGTG-3′	
ABCA9	
forward primer:5′-CAGAGGGAGTGAAGAGAAAGC-3′,	
reverse primer: 5′-GCTCTGTGTTTGTGAAAGTGG-3′	
CXorf57:	
forward primer:5′-GCAGTATAGGGAACAAAAGCG-3′,	
reverse primer: 5′-TGCTTGAGATGTTGAGGGAC-3′	
GPR22:	
forward primer:5′-CCACTGTCATACCCACTAAGC-3′,	
reverse primer: 5′-ATGCAGTAAAGTACCAGGACG-3′	
STXBP5L:	
forward primer:5′-GATCAAGTGACCTGTACCAGC-3′,	
reverse primer: 5′-ATTTACATGGTCTGAGGTGGG-3′	
MSANTD4	
forward primer:5′-CAGAGGTCAAAGTGGAAGAGG-3′,	
reverse primer: 5′-ATCAATGTGAGGGAAGTCAGG-3′	
RRP15	
forward primer:5′-GAAATGCTGTGCAGAGTGAAG-3′,	
reverse primer: 5′-TCCTGCTTCCTTAACCTTTTCG-3′	
UGT1A2	
forward primer:5′-TCTGCGTTCTCTTTCCTGTG-3′,	
reverse primer: 5′-AGCATGTTCTGGACCCTTTG-3′	
IRF4	
forward primer:5′-AACAAGCTAGAAAGAGACCAGAC-3′,	
reverse primer: 5′-TCACCAAAGCACAGAGTCAC-3′	
TFAP2D	
forward primer:5′- AAAGATGATCCTAGCCACCAAG-3′,	
reverse primer: 5′-TGTGTTAAGTGCCTCTGGATG-3′	
TRHDE	
forward primer:5′-AGGAAGGCTTTGCTCACTAC-3′,	
reverse primer: 5′-CTGTGATACTGGATGGGAACTG-3′	
ASMT	
forward primer:5′-GAAGTGGGACAGGAAGTGAG-3′,	
reverse primer: 5′-CGGGAACAGGAAGTGGC-3′	
CAPS	
forward primer:5′-AGCTCGAAGACACAATCCG-3′,	
reverse primer: 5′-TCCATGTCCACTGCAAAGAG-3′	
COMMD10	
forward primer:5′-AGTGGGATGGCAGCTTAAC-3′,	
reverse primer: 5′-TCGAACAGCTCCTTGTGATTG-3′	
VSTM4	
forward primer:5′-CCTGGCAGTCTGTGTTTCA-3′,	
reverse primer: 5′-CTCTTACCCTTCTGTGGCTG-3′	
FBXO9	
forward primer:5′-ATGAGAGTCCGGCTGAGAGA-3′,	
reverse primer: 5′-AGAGCTTCTTCCTGCTCTGC-3′	
18s	
forward primer:5′-CTCAACACGGGAAACCTCAC-3′,	
reverse primer: 5′-CGCTCCACCAACTAAGAACG-3′	

Western blotting analysis

BMDMs-induced osteoclast (1 × 105 cells/well in 12-well plates) were treated with or without ZA (50 μM) for 48 h, the total protein was isolated using RIPA Buffer (Solarbio, Beijing, China), and total protein concentration was quantified by BCA protein assay kit (Beyotime). A total of 20 μg of protein were separated by 8% SDS-PAGE and transferred to PVDF membranes (Merck Millipore, Billerica, MA, USA). After blocking with 5% bovine serum albumin or 5% nonfat milk, the membranes were incubated with anti-FBXO9 (1:1,000, PA5-25475; Thermo Fisher Scientific, Waltham, MA, USA), p53 (1.0 µg/mL, MA5-14067; Thermo Fisher Scientific, Waltham, MA, USA), ubiquitin (1:2,000, ab134953; Abcam, Cambridge, England), and GAPDH (1:5,000, MA1-16757; Thermo Fisher Scientific, Waltham, MA, USA) overnight at 4 °C. Then the membranes were treated with HRP-conjugated anti-mouse or rabbit secondary antibody (1:5,000) for 1 h at room temperature.

Co-immunoprecipitation

BMDMs-induced osteoclast were lysed using NP40 buffer (10 mM Tris-HCl at pH 8.0, 140 mM NaCl, 1.5 mM MgCl2, 0.5% Nonidet-P40, 20 mM dithiothreitol, 500 U/mL RNAsin, and 0.5% [w/v] deoxycholate), cell lysates were incubated with FBXO9 (2 µg/ml; PA5-25475) or p53 (2 µg/ml; MA5-14067) antibody for 4 h, then Protein A/G beads (Thermo Fisher Scientific, Waltham, MA, USA) were added to the IP reactions and left rotating overnight at 4 °C, then beads were washed by PBS containing protein inhibitors for three times, then the immunoprecipitates were analyzed using western blotting with FBXO9 antibody or p53 antibody.

Statistical analysis

All the data are shown as the mean ± standard error of mean (SEM) from three independent experiments. Statistical analysis was carried out using SPSS 19.0 software (IBM Corp., New York, USA). The significance between two groups was analyzed using one-way ANOVA followed by Tukey-Kramer multiple comparisons test or unpaired Student’s t-test. p < 0.05 was considered to indicate a statistically significant difference.

Results

ZA treatment facilitated the ferroptosis of osteoclasts

To investigate the function of ZA on osteoclasts, RAW264.7 cells and bone marrow-derived macrophages (BMDMs) were pre-treated with RANKL for 6 days followed by ZA treatment in different concentrations (5, 10, and 50 uM) for 48 h, the cell model was confirmed by TRAP staining (Figs. 1A and 1B). Cell viability was analyzed by CCK-8 assay. As shown in Figs. 1C and 1D, ZA treatment suppressed cell viability in a dose-dependent manner. The results from FDA staining also showed that the effect of ZA on promoting cell viability of osteoclasts (Figs. 1E and 1F). Impressively, the CCK-8 assay results showed that cell death of osteoclasts induced by ZA treatment was obviously blocked by ferrostatin-1 (Fer-1, a specific inhibitor of ferroptosis) but not necrostatin-1 (a specific inhibitor of necroptosis) and ZVAD-FMK (a specific inhibitor of apoptosis) (Figs. 1G and 1H).

Figure 1 ZA treatment facilitated the ferroptosis of osteoclasts.

The osteoclasts cell model induced by RANKL (50 ng/ml) treatment. (A and B) Multinucleated cells were visualized by tartrate-resistant acid phosphatase (TRAP) staining. (C and D) Cell viability of Raw264.7 and BMDM derived osteoclasts was assessed using CCK8 assay after treatment with different concentrations of ZA (5, 10, and 50 μM) (n = 3). (E and F) Cell viability of Raw264.7 and BMDM derived osteoclasts was assessed using FDA staining after treatment with different concentrations of ZA (5, 10, and 50 μM) (n = 3). (G and H) Cell viability of Raw264.7 and BMDM derived osteoclasts was assessed using CCK8 assay after treatment with ZA for 48 h (50 μM) in the presence or absence of 10 μM of ZVAD-FMK, 2 μM of Fer-1, or 10 μM of necrostatin-1 (n = 3). *p < 0.05, **p < 0.01.

To define the role of ZA in the ferroptosis of osteoclasts, the ferroptosis signaling was evaluated in osteoclasts after ZA treatment. As shown in Figs. 2A–2C and 2F–2H, ZA treatment markedly increased the Fe2+ level, MDA content, and ROS level in a dose-dependent manner in osteoclasts, differentiated from RAW264.7 cells and BMDMs, suggesting the promotion of ferroptosis signaling in osteoclasts treated by ZA. Besides, ZA treatment also suppressed the levels of Gpx4 and GSH in a dose-dependent manner in osteoclasts (Figs. 2D, 2E, 2I, and 2J). These results demonstrate that ZA treatment facilitates the ferroptosis of osteoclasts.

Figure 2 ZA treatment facilitated the ferroptosis of osteoclasts.

(A–E) The level of Fe2+, MDA content, ROS level, the level of Gpx4, and GSH content in Raw264.7 derived osteoclasts was assessed by Elisa assay after treatment with different concentrations of ZA (5, 10, and 50 μM) (n = 3) *p < 0.05, **p < 0.01, ***p < 0.001. (F–J) The level of Fe2+, MDA content, ROS level, the level of Gpx4, and GSH content in BMDM derived osteoclasts was assessed by Elisa assay after treatment with different concentrations of ZA (5, 10, and 50 μM) (n = 3) *p < 0.05, **p < 0.01, ***p < 0.001.

FBXO9 was downregulated in osteoclasts after ZA treatment

To investigate the mechanism underlying ZA-induced osteoclasts ferroptosis, the differentially expressed genes (DEGs) of osteoclasts induced by bisphosphonates alendronate- and risedronate-treatment were obtained from GSE63009, and the common DEGs were identified by venn diagram analysis. As shown in Fig. 3A, 18 common genes were identified (CFAP53, COL14A1, ARSJ, ABCA9, CXorf57, GPR22, STXBP5L, MSANTD4, RRP15, UGT1A2, IRF4, TFAP2D, TRHDE, ASMT, CAPS, COMMD10, VSTM4, FBXO9). The level of these 18 genes was evaluated in osteoclasts treated with or without ZA using qPCR analysis. Figures 3B and 3C showed that only FBXO9 was significantly decreased in osteoclasts after treatment with ZA. Similar to the qPCR results, the results from western blotting showed that the expression of FBXO9 was obviously decreased in osteoclasts after treatment with ZA. These results indicate that the FBXO9 was downregulated by ZA treatment (Fig. 3D).

Figure 3 FBXO9 was downregulated in osteoclasts after ZA treatment.

(A) Venn analysis of DEGs of alendronate and risedronate-treated osteoclast. (B and C) The mRNA level of 18 genes in Raw264.7 and BMDM derived osteoclasts was assessed using qPCR after treatment with ZA (50 μM) (n = 3) *p < 0.05, **p < 0.01. (D) The protein level of FBXO9 in Raw264.7 and BMDM derived osteoclasts was assessed using western blot after treatment with ZA (50 μM).

FBXO9 inhibition facilitated the ferroptosis of osteoclasts

To investigate the function of FBXO9 on osteoclasts, the expression of FBXO9 was down-regulated by si-FBXO9 in BMDMs-differentiated osteoclasts. As shown in Figs. 4A and 4B, the expression of FBXO9 was significantly decreased by si-FBXO9. Cell viability of osteoclasts was significantly decreased in the FBXO9 knockdown group compared with the control group (Fig. 4C). The results from FDA staining also showed the effect of FBXO9 on inhibiting cell viability (Figs. 4D and 4E).

Figure 4 FBXO9 inhibition facilitated the ferroptosis of osteoclasts.

(A) The mRNA level of FBXO9 in BMDM derived osteoclasts was assessed using qPCR after treatment with or without si-FBXO9 (n = 3). ***p < 0.001. (B) The protein level of FBXO9 in BMDM derived osteoclasts was assessed using western blot after treatment with or with out si-FBXO9 (n = 3). **p < 0.01. (C) Cell viability of BMDM derived osteoclasts was assessed using CCK8 assay after treatment with or without si-FBXO9 (n = 3). **p < 0.01. (D and E) Cell viability of BMDM derived osteoclasts was assessed using FDA staining after treatment with or without si-FBXO9 (n = 3). *p < 0.05. (F–J) The level of Fe2+, MDA content, ROS level, the level of Gpx4, and GSH content in BMDM derived osteoclasts was assessed by Elisa assay after treatment with or without si-FBXO9 (n = 3) *p < 0.05, **p < 0.01.

To investigate the role of FBXO9 in ferroptosis of osteoclasts, the ferroptosis signaling was evaluated in osteoclasts differentiated from BMDMs. As shown in Figs. 4F–4J, FBXO9 knockdown significantly increased the Fe2+ level, MDA content, ROS level and decreased the GPX4 level, GSH content in osteoclasts. These results suggested that FBXO9 inhibition facilitates the ferroptosis of osteoclasts.

ZA treatment facilitated the ferroptosis of osteoclasts by suppressing FBXO9

To explore whether FBXO9 mediated the function of ZA in regulating the ferroptosis of osteoclasts, the osteoclasts differentiated from BMDMs were treated by ZA in the presence or absence of FBXO9. As shown in Figs. 5A and 5B, the qPCR and western blotting analysis results showed that the expressions of FBXO9 were decreased by ZA treatment and restored by FBXO9 overexpression. CCK8 results showed that cell viability of osteoclasts was decreased by ZA treatment, but these effects were blocked by FBXO9 overexpression (Fig. 5C). Consistently, the FDA staining also showed the inhibition of ZA on osteoclasts cell viability was restored by FBXO9 overexpression (Figs. 5D and 5E). Besides, the Fe2+ level, MDA content, and ROS level were obviously increased and the GPX4 level, GSH content was significantly decreased by ZA treatment, while these effects were blocked by FBXO9 overexpression (Figs. 5F–5J). These results suggested that ZA treatment facilitates the ferroptosis of osteoclasts by suppressing FBXO9.

Figure 5 ZA treatment facilitated the ferroptosis of osteoclasts by suppressing FBXO9.

(A) The mRNA level of FBXO9 in BMDM derived osteoclasts was assessed using qPCR after treatment with ZA (50 μM) in the presence or absence of FBXO9 (n = 3). **p < 0.01. (B) The protein level of FBXO9 in BMDM derived osteoclasts was assessed using western blot after treatment with ZA (50 μM) in the presence or absence of FBXO9 (n = 3). *p < 0.05. (C) Cell viability of BMDM derived osteoclasts was assessed using CCK8 assay after treatment with ZA (50 μM) in the presence or absence of FBXO9 (n = 3). *p < 0.05, **p < 0.01. (D and E) Cell viability of BMDM derived osteoclasts was assessed using FDA staining after treatment with ZA (50 μM) in the presence or absence of FBXO9 (n = 3). *p < 0.05. (F–J) The level of Fe2+, MDA content, ROS level, the level of Gpx4, and GSH content in BMDM derived osteoclasts was assessed by Elisa assay after treatment with ZA (50 μM) in the presence or absence of FBXO9 (n = 3) *p < 0.05, **p < 0.01, ***p < 0.001.

FBXO9 inhibition facilitated the ferroptosis of osteoclasts by blocking the ubiquitin mediated-proteasome degradation of p53

Previous studies showed that FBXO9, an E3 ubiquitin ligase, mediated protein stability through ubiquitin mediated-proteasome degradation (Liu et al., 2020). Given that dysregulated ubiquitination has been widely reported to be involved in many diseases by regulating cell ferroptosis (Yang et al., 2020; Zhang et al., 2020), the target of FBXO9 was predicted by ubibrowser (http://ubibrowser.ncpsb.org.cn/ubibrowser/). Figure 6A showed the 20 potential target genes that interacted with FBXO9. Interestingly, among these genes, the p53 gene is an important regulator of ferroptosis. We next explored whether FBXO9 decreases p53 protein level by promoting its ubiquitination-mediated degradation. Knockdown of FBXO9 in osteoclasts did not change the p53 mRNA level (Fig. 6B). Fascinatingly, the protein level of p53 was significantly increased after FBXO9 inhibition (Fig. 6C), suggesting that FBXO9 decreased p53 expression possibly by the ubiquitin-proteasome-mediated degradation. Next, a reciprocal Co-IP assay was performed to confirm whether FBXO9 directly interacts with p53. As shown in Fig. 6D, a positive p53 signal was observed in the protein complex pulled down by FBXO9 antibody. Meanwhile, FBXO9 was also detected in the co-immunoprecipitation complex pulled-down by p53 antibody. Next, cycloheximide assay (CHX) was performed to detect the protein stability of p53 in osteoclasts transfected with si-FBXO9. As shown in Fig. 6E, the protein stability of p53 was obviously increased in the FBXO9 knockdown osteoclasts. Then the p53 ubiquitination was assessed through IP with FBXO9 antibody and subsequent western blotting with ubiquitin antibody. Figure 6F showed that FBXO9 knockdown obviously decreased p53 ubiquitination in osteoclasts (Fig. 6F). In conclusion, these results indicated that FBXO9 directly interacts with p53 and promotes its degradation.

Figure 6 FBXO9 inhibition facilitated the ferroptosis of osteoclasts by blocking the ubiquitin mediated-proteasome degradation of p53.

(A) The target of FBXO9 was predicted by ubibrowser. (B) The p53 mRNA expression in the FBXO9 knockdown and control cell was assessed by qPCR (n = 3). (C) The protein level of p53 in the FBXO9 knockdown and control cell was assessed by western blot (n = 3). (D) FBXO9 directly interacts with p53. The proteins from BMDM derived osteoclasts were IP with IgG or antibodies against FBXO9 and p53, following by western blot analysis (n = 3). (E) The stability of p53 protein was regulated by FBXO9. BMDM derived osteoclasts treated with or without si-FBXO9 in the presence of cycloheximide (CHX, 25 ug/ml) for various times as indicated and cell lysates were then assessed by western blot (n = 3). **p < 0.01. (F) The cell lysates isolated from scramble and si-FBXO9 infected BMDM derived osteoclasts were immunoprecipitated with anti-p53 antibody, then analyzed by western blot using ubiquitin antibody (n = 3).

Discussion

BRONJ is one of the severe complications of BPs administration reported by Marx (2003). It usually occurs in patients with bone metastatic cancer or osteoporosis, and undergoes bisphosphonate therapy. ZA is a kind of nitrogen-containing bisphosphonates and is widely used in the treatment of bone metastatic cancer and osteoporosis. Zhu et al. (2019) reported that ZA facilitates TLR-4-mediated M1 type macrophage polarization in the development of BRONJ. Huang et al. (2019) demonstrated that ZA inhibited osteoclast differentiation and function by regulating the NF-κB and JNK signaling pathways. However, the mechanisms underlying ZA regulates osteoclast function in the occurrence of BRONJ remains unclear. In the current study, we clarified that ZA promotes osteoclasts ferroptosis by inhibiting FBXO9-mediated p53 ubiquitination and degradation, as evidence by (I) ZA treatment facilitated the ferroptosis of osteoclasts; (II) FBXO9 was downregulated in osteoclasts after ZA treatment; (III) FBXO9 inhibition facilitated the ferroptosis of osteoclasts; (IV) ZA treatment facilitated the ferroptosis of osteoclasts by suppressing FBXO9; (V) FBXO9 inhibition facilitated the ferroptosis of osteoclasts by blocking the ubiquitin mediated-proteasome degradation of p53.

Although a growing body of research have explored the role of BPs in the pathogenesis of BRONJ, the mechanism of BPs on the development of BRONJ is not completely understood. Growing studies have demonstrated that BPs have high affinity to hydroxyapatite crystals, thereby suppressing the osteoclasts resorptive ability by inducing the apoptosis of osteoclasts (Favia et al., 2018; Russell, 2011). Moreover, due to the lack of cytokines released by osteoclasts, the differentiation of osteoblasts was blocked, thus suppressing the healing ability of bone, suggesting that the differentiation of osteoclasts plays an important role in the development of BRONJ (AlDhalaan, BaQais & Al-Omar, 2020). More recently, ZA has been reported to inhibit osteoclast differentiation by regulating the NF-κB and JNK signaling pathways (Huang et al., 2019). Another study has shown that ZA inhibits osteoclast differentiation by interrupting RANKL/RANK pathway (Li et al., 2019). Consistent with previous studies, we confirmed that ZA decreased cell viability of osteoclasts induced by RANKL, specifically ZA-induced cell viability decrease was blocked by ferroptosis inhibitor, suggesting an important role of ferroptosis in the development of BRONJ.

Ferroptosis is a kind of iron- and ROS-dependent form of cell death, different with necrosis, apoptosis, and other forms of cell death. Right now, almost all the mechanisms of ferroptosis are associated with reactive oxygen species (ROS) (Li et al., 2020). Given the role of ZA in regulating ROS production (Sehitoglu et al., 2015; Wang et al., 2018; Liu et al., 2021), here we investigated whether ZA suppresses the growth of osteoclast by accelerating ferroptosis. We found that the ferroptosis-related marker such as the levels of Fe2+, MDA content, ROS level was obviously increased in the osteoclasts treated with ZA, suggesting the ZA induced the ferroptosis of osteoclasts. However, the underlying mechanism of ZA-induced osteoclast ferroptosis reminds unknown.

To elucidate the mechanism of ZA-induced osteoclast ferroptosis, we compared the expression profiles of osteoclasts in the presence or absence of alendronate or risedronate treatment, and got 18 genes with significant differences in the osteoclasts treated by BPs. Among these18 genes, FBXO9 was identified to be significantly reduced in ZA-treated osteoclasts. Further experiment showed that FBXO9 knockdown promoted the ferroptosis of osteoclasts, and the ferroptosis of osteocalsts induced by ZA was blocked by FBXO9 overexpression, suggesting that ZA promotes the ferroptosis of osteoclasts by downregulating the expression of FBXO9.

The F-box only protein 9 (FBXO9), a member of the F-box protein family, is the substrate recognition subunit of skp1-cullin1-f-box E3 ligase complex and plays a key role in ubiquitination and subsequent target protein degradation (Lee et al., 2016). Liu et al. (2020) demonstrated that FBXO9 interacted with Neurog2 and promoted its destabilization is a major contributor in directing multipotent NC progenitors toward glial lineage. Vanesa Fernández-Sáiz et al. (2013) demonstrated that, under the growth factor deprivation condition, FBXO9-mediated ubiquitination of Tel2 and Tti1 inactivated mTORC1, but activated the PI3K/Akt pathway to increase survival of multiple myeloma. However, the function of FBXO9 in the development of BRONJ and the regulatory mechanism remain unclear. Growing studies suggested that E3 ubiquitin ligase regulates ferroptosis by degrading substrates. Yang et al. (2020) reported that Nedd4 ubiquitylates VDAC2/3 to suppress erastin-induced ferroptosis in melanoma. Another study showed that TRIM26 facilitates the ferroptosis of HSCs to suppress liver fibrosis by mediating the ubiquitination of SLC7A11 (Zhu et al., 2021). Therefore, we speculated whether FBXO9 also regulates ferroptosis by mediating the ubiquitination of target genes. Interestingly, we found that p53, a key upstream regulator of ferroptosis, is one of the FBXO9 targets. Our data showed that FBXO9-knowdown did not change the p53 mRNA level but significantly increased the p53 protein level, suggesting that FBXO9-mediated p53 expression by the ubiquitin-proteasome system. Further experiment showed that FBXO9 directly interacts with p53 and the ubiquitination level of p53 was downregulated by FBXO9 knockdown. In addition, the protein stability of p53 was promoted by FBXO9 knockdown. These data suggesting that p53 is the direct target of FBXO9 and FBXO9-mediated p53 ubiquitination and degradation in osteoclast.

Conclusions

Taken together, the current data demonstrated that FBXO9 was downregulated in ZA-treated osteoclast and promoted osteoclasts ferroptosis by inhibiting FBXO9-mediated p53 ubiquitination and degradation. Our study provided a possible theoretical target for the clinical treatment of BRONJ.

There are still some deficiencies in the current research, such as the current conclusions still need to be further confirmed by clinical and animal experiments.

Supplemental Information

Supplemental Information 1 Raw data for non-WB.

Click here for additional data file.

Supplemental Information 2 Raw data for WB.

Click here for additional data file.

Additional Information and Declarations

Competing Interests

Author Contributions

Data Availability

The authors declare that they have no competing interests.

Xingzhou Qu performed the experiments, prepared figures and/or tables, and approved the final draft.

Zhaoqi Sun analyzed the data, authored or reviewed drafts of the paper, and approved the final draft.

Yang Wang conceived and designed the experiments, authored or reviewed drafts of the paper, and approved the final draft.

Hui Shan Ong conceived and designed the experiments, prepared figures and/or tables, and approved the final draft.

The following information was supplied regarding data availability:

The raw data is available in the Supplemental Files.

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
