# Peer review of "Zoledronic acid promotes osteoclasts ferroptosis by inhibiting FBXO9-mediated p53 ubiquitination and degradation"

_PeerJ, doi:10.7717/peerj.12510_

## Round 0.1 · original submission · Major Revisions

Please carefully address all reviewers' concerns and request assistance with English language before submitting the revised manuscript.

Reviewer 1 ·

Basic reporting

Qu et al. described in their manuscript that zoledronic acid facilitates osteoclast ferroptosis. They concluded that this promoted osteoclast ferroptosis is caused by downregulation of FBXO9 expression in osteoclasts, so that FBXO9-mediated degradation of p53 is blocked. They claimed, “for the first time, we showed that ZA promotes the ferroptosis of osteoclast by increasing the protein stability of p53” (line 84-85). However, I am not convinced by the conclusion drawn by the authors because of following issues:
Major comments:
1. This manuscript is not written in sound English.
2. The authors need to make their writing precisely and clearly: the authors must provide precise information of materials (such as catalog number for the bone marrow cells from ATCC); the authors must replace “we found that ZA treatment suppressed (the) cell viability of osteoclasts” (in Abstract, and line 30-31, line 276-277) with “we confirmed…” as this is not their original finding; the authors should use science terms such as “osteoclasts” but not “osteoclasts cells”, etc.; the authors should provide the full name for any abbreviations used, such as the full name for FDA (fluorescein diacetate);
3. The authors should avoid false information, such as “osteoclasts proliferation” in line 63. Osteoclast does not proliferate.
4. The authors must provide evidence (e.g., osteoclast-specific marker staining) that osteoclast-like cells were successfully generated by them from both RAW264.7 cell line and the primary cells in vitro. Although cells used in this study were RANKL-treated, or RANKL plus M-CSF treated, I am not convinced that osteoclastogenesis was successfully induced. In other words, there is no evidence in this manuscript showing multinucleated cell formation. (Such as cells shown in Figure 4D).
5. Even osteoclastogenesis was succussed, cells studied in this manuscript were a mixture of osteoclasts and mononucleated cells (pre-fusion osteoclasts), not osteoclasts as the authors described. There is no cell separation (separate multinucleated cells from mononucleated cells) performed in the study.
Minor comments:
1. Names for all authors were marked with “1”. This mark should be removed because they are affiliated to a same institute.
2. Instead of Hotmail and 163, better use academic email address (institutional address) as the corresponding author’s contact address.
3. I assume that the authors used RANKL alone, and RANKL plus M-CSF (line 97-98) to turn RAW264.7 cells and BMDMs to osteoclasts in vitro, respectively. The authors should clarify this.
4. Better combine those two paragraphs in line 91-101, because the two paragraphs all talked about in vitro osteoclastogenesis.
5. Two approaches were used to determine cell viability. The two paragraphs in line 102-115 for this objective can be combined. “The cell viability” in the entire paper should be replaced with “cell viability”.
6. The authors better describe their methods briefly, instead of simply saying “as the manufacturer’s instructions” (line 119, etc.)
7. Move line 73-88 to Discussion.
8. It is unclear at what differentiation stage that the cells were transfected by the pcDNA-FBXO9 or si-FBXO9 RNA.
9. The RT-qPCR primer sequences for 18s should be provided.
10. It should be described as cell number per well, not simply cell number (such as line 123: 5x100,000 should be 5x100,000/well; line 149: 1x100,000 should be 1x100,000/well).
11. Line 150-151, “the concentration was quantified” should be “total protein concentration was quantified”.
12. Exact amount of total cell lysates (reagents) should be used for the Western blot analysis (and other assays). “Approximately” in line 152 is improper.
13. Buffer formulation for NP40 buffer (line 160) should be provided.
14. Line 279-286 in the Discussion section sounds like Introduction.
15. Figures are not fully discussed.

Experimental design

See in "Basic reporting".

Validity of the findings

See in "Basic reporting".

Reviewer 2 ·

Basic reporting

no comment

Experimental design

1.It's better to test this mechanisms in in vivo or animal models to make this hypothesis more convincing.
2.in addition, authors should explain more specified about why ZA induced osteoclasts ferroptosis instead of apotosis or autophagy mechanisms.
3.Is FBXO9-related pathway lead to ferroptosis?

Validity of the findings

this study uncovered the important role of ferroptosis in ZA induced osteoclasts apotosis. these results will help to elucidate the mechanism of BRONJ. this study could be considered to be published.

Additional comments

no comment

Reviewer 3 ·

Basic reporting

In this study, the authors showed that ZOL induced osteoclast death through ferroptosis. For the stimulation of the ferroptosis, ZOL inhibited FBXO9 expressions, which facilitate the ubiquitination of p53.
In general, the manuscript writing is clear with proficient English. However, a few spelling and spacing should be thoroughly checked.
- Introduction and references are proper.
- Figures are relevant to the content.

Experimental design

The experiments were well-designed and conducted in order to prove the objective. A few points are needed to be revised before accepting.
Materials and methods
1. Line 134 check the sentence ending
2. Line 149 and 160 add more detail on what cell type used for the induction of osteoclasts
3. Line 152 “Approximately 20 μg of l protein were separated….” Spelling check on protein
4. Line 154 “the membranes were incubated with an-FBXO9…” the author change an-FBXO9 to anti-FBXO9

Validity of the findings

The article will add to the existing knowledge pool and could be applied for the future use.

I would suggest using SD instead of SEM for the statistical analysis.

Discussion
1. The authors should add the clinical implication of this study in terms of treatment or management of BRONJ.
2. What should be the future study?

Conclusion is too short.

---

## Round 0.2 · Minor Revisions

I am attaching the file where I have annotated in lines 33 and 208 necessary changes including what you had been requested to change by one reviewer which was addressed in the rebuttal letter but not in the text.

---

## Round 0.3 · Minor Revisions

I am attaching the PDF of your tracked revised manuscript where in line 31 I highlighted the verb that you need to use ( "confirmed") as requested by one reviewer. I had already requested this change, so please do comply.

---

## Round 0.4 · Minor Revisions

Please correct as previously requested. By mistake you uploaded again the previous version. I am attaching again a PDF with the correction requested.

---

## Round 0.5 · accepted · Accept

Thank you for addressing the concerns of the reviewers.